# HoloGarment: 360° Novel View Synthesis of In-the-Wild Garments

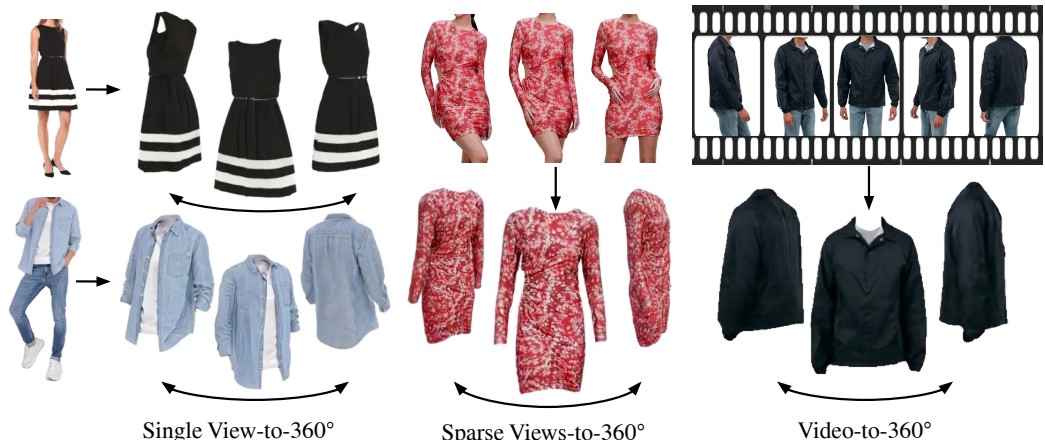

Single View-to-360°          Sparse Views-to-360°          Video-to-360°

Figure 1: HoloGarment enables 360° novel view synthesis of real-world garments in images and videos.

## Abstract

Novel view synthesis (NVS) of in-the-wild garments is a challenging task due significant occlusions, complex human poses, and cloth deformations. Prior methods rely on synthetic 3D training data consisting of mostly unoccluded and static objects, leading to poor generalization on real-world clothing. In this paper, we propose HoloGarment (**Holo**gram-**Garment**), a method that takes 1-3 images or a continuous video of a person wearing a garment and generates 360° novel views of the garment in a canonical pose. Our key insight is to bridge the domain gap between real and synthetic data with a novel implicit training paradigm leveraging a combination of large-scale real video data and small-scale synthetic 3D data to optimize a shared garment embedding space. During inference, the shared embedding space further enables dynamic video-to-360° NVS through the construction of a garment "atlas" representation by finetuning a garment embedding on a specific real-world video. The atlas captures garment-specific geometry and texture across all viewpoints, independent of body pose or motion. Extensive experiments show that HoloGarment achieves state-of-the-art performance on NVS of in-the-wild garments from images and videos. Notably, our method robustly handles challenging real-world artifacts – such as wrinkling, pose variation, and occlusion – while maintaining photorealism, view consistency, fine texture details, and accurate geometry.

## 1 Introduction

The rise in online retail, virtual try-on, and digital fashion design is driving the demand for high-quality digital garment visualizations. While images and videos offer limited, pose-dependent views of a garment, they fall short of providing a **full 360° garment representation – one that is free of occlusions and wrinkles**. Furthermore, manually capturing such 360° views is highly impractical and costly. Therefore, there is strong interest in discovering an automatic method to generate high-quality, high-fidelity novel views of garments from images and videos.

This is a challenging task, since real-world garments are inherently complex, containing **deformations, occlusions, and pose variations** when worn. Existing methods for novel view synthesis

(NVS) of general objects (Gao et al., 2024; Xiang et al., 2024) only handle a fixed number of input views and are constrained to **static and unoccluded objects**. As a result, these approaches perform poorly on real-world garments and do not handle an arbitrary number of input views, such as from a video. While some methods handle finetuning on video (Wang et al., 2019; Zakharov et al., 2019), they are prone to overfitting to the shape and appearance of the subject in the input frames, thus failing to generate unoccluded, static novel views of a standalone garment in a dynamic video.

Another challenge is the lack of diverse, realistic 3D garment datasets. While 2D garment data (images and videos) are abundantly available online, they are missing ground-truth 3D representations. As such, past works (Bang et al., 2021; Gao et al., 2023; He et al., 2024; Korosteleva & Lee, 2022; Lim et al., 2023; Richardson et al., 2023; Sarafianos et al., 2024; Su et al., 2020) often leverage purely synthetic garment data (He et al., 2024; Li et al., 2023; Zhu et al., 2020), but tend to overfit to simplistic shapes and patterns and generalize poorly to diverse garment shapes and textures.

To overcome these limitations, we ask: (1) Can we train a real-world garment NVS model by leveraging abundant real-world 2D data, even in the absence of paired ground-truth 3D assets? (2) In contrast to sparse-frame conditioning, can we leverage dense frames from a dynamic video sequence to learn a robust and high-fidelity garment representation for NVS?

In this paper, we propose **HoloGarment, a video diffusion model for garment NVS from images and videos of in-the-wild dressed humans**. Our key insight is **a novel implicit training paradigm**, where two or more distinct training tasks indirectly train a model to perform the target task for which ground truth data is not available. By implicitly training with a combination of real 2D data and synthetic 3D assets, our method learns a shared garment embedding space between both domains that enables real-world garment novel view synthesis. In doing so, we bypass the limitations of synthetic-only 3D datasets to handle challenging real-world garment images and videos. Furthermore, **we introduce the notion of a garment "atlas"**, a finetuned garment embedding optimized on a specific dynamic video featuring a person wearing the garment. The "atlas" bridges the gap between finetuning (2D) and inference (3D) modalities, **enabling the novel task of *video*-to-NVS generation**, as well as eliminates the need for arbitrary input view selection. Our experiments showcase HoloGarment's capability to generate high-quality, high-fidelity 360° novel views across a variety of garment types, including tops, dresses, jackets, rompers, and pants, even those containing occlusions, pose variations, and deformations. We further quantitatively and quantitatively demonstrate that our method **achieves state-of-the-art results** compared to related methods.

## 2 RELATED WORK

**Novel View Synthesis with Diffusion Models:** Novel view synthesis refers to generating novel object views from limited observations. Many diffusion-based NVS methods use 3D datasets to fine-tune a pretrained text-to-image diffusion model (Gao et al., 2024; Liu et al., 2023b;c; Shi et al., 2023a;b; Wang & Shi, 2023) or video diffusion model (Kwak et al., 2023; Wang et al., 2023; Zhou et al., 2025). However, their reliance on 3D data limits their ability to handle real-world objects effectively. Several works have explored using 2D diffusion priors to enhance 3D consistency (Lin et al., 2023; Poole et al., 2022; Shi et al., 2023b), but do not tackle cases where input views contain incomplete information (i.e. occlusions) or inconsistencies (i.e. deformations, pose changes). Existing NVS methods therefore struggle with diverse garments in complex, dynamic real-world scenarios. In this work, we address this challenging case directly by training a video diffusion model implicitly on real 2D videos and synthetic 3D assets, enabling our method to robustly generate consistent novel views of real-world garments.

**3D Garment Reconstruction:** Related to the task of garment NVS is 3D garment reconstruction, which aims to recover the 3D geometry of a garment in an image. One avenue of methods explores 2D sewing pattern estimation (Bang et al., 2021; He et al., 2024; Korosteleva & Lee, 2022; Lim et al., 2023; Liu et al., 2023a), which predicts flat patterns that can be draped onto the person's 3D structure, but neglects to preserve texture details. Other recent methods focus on texture estimation by utilizing template garment meshes to achieve better realism in garment representation (Gao et al., 2023; Richardson et al., 2023; Sarafianos et al., 2024; Su et al., 2020). A major limitation of both approaches is their reliance on limited synthetic 3D garment datasets, including DressCode (He et al., 2024) and GarverseLOD (Luo et al., 2024), which lack ground-truth textures. As a result, these

methods do not generalize well to real-world garment inputs. In contrast, our approach eliminates the reliance on purely synthetic data, input meshes, and complex templates.

**Subject-Specific Finetuning:** Subject-specific finetuning, like DreamBooth (Ruiz et al., 2023) and similar video-specific methods (Wang et al., 2019; Zakharov et al., 2019), customize a pretrained generative model to a specific subject. While this paradigm has been extended for human identity preservation (Karras et al., 2023; Zhu et al., 2024), it has not yet been applied to garment identity specifically, which comes with unique challenges. One specific limitation is that current subject-finetuning methods tend to overfit to the pose and shape of the target subject. As such, existing methods will replicate the motion, occlusions, deformations, and wrinkling of a dynamic garment video that is used for finetuning. This is undesirable for synthesizing a static video of a dynamic garment in an unoccluded, canonical A-pose. We address this by explicitly disentangling animation and spin motion via split temporal blocks in our network, while still sharing the same garment appearance encoder. As a result, we are able to finetune a garment-specific embedding on a dynamic garment video and still generate novel views of the garment in a static pose, without overfitting to the original dynamic motion.

## 3 PRELIMINARIES

**Diffusion Models:** Diffusion models are a class of generative models capable of synthesizing high-fidelity data, particularly images and videos (Dhariwal & Nichol, 2021; Ho et al., 2020; Sohl-Dickstein et al., 2015; Song et al., 2020; Song & Ermon, 2019). In the forward process, the data is transformed incrementally into pure Gaussian noise over a discrete number of steps. Then, a diffusion model (typically a UNet) is trained to predict the reverse process, which iteratively denoises the Gaussian noise back into a clean data sample. To be precise, at timestep $t$, diffusion model $\epsilon_\theta$ with parameters $\theta$ predicts noise $\hat{\epsilon}_t$ added to the noisy data sample $z_t$. With conditioning signal(s) $c$, one diffusion timestep is defined as: $\hat{\epsilon}_t = \epsilon_\theta(z_t, t, c)$. From the predicted noise, the denoised data sample $\hat{z}_{t-1}$ can be estimated. The diffusion model is optimized by the following objective function:

$$\mathcal{L} = ||\epsilon_t - \epsilon_\theta(z_t, t, c)||_2^2 \tag{1}$$

**Video Diffusion Transformer Models:** While conventional diffusion models often leverage a UNet backbone, the diffusion transformer (DiT) (Peebles & Xie, 2022) model replaces this with a Transformer architecture, leading to superior scalability and performance. Fashion-VDM (Karras et al., 2024) extends DiT into a video model with temporal blocks (e.g. 3D convolutions and temporal attention layers) and progressive temporal training. Paired with parallel UNet encoders to disentangle person and garment conditioning signals (Zhu et al., 2023), Fashion-VDM achieves superior performance for video try-on.

## 4 METHOD

Given 1-3 images $I_g$ of a garment $g$ and a driving pose sequence $J = (J_{2D}, J_{3D})$ represented in 2D and 3D, HoloGarment generates novel garment views $\hat{V}_g$ following the driving poses. In this section, we introduce our model architecture (4.1) and implicit training strategy with real 2D and synthetic 3D garment data (4.2). Then, we describe how this unlocks image(s)-to-360° NVS (4.3), as well as video-to-360° NVS capabilities via finetuning a garment "atlas" (4.4).

### 4.1 HOLOGARMENT

At its core, HoloGarment consists of an image- and pose-conditioned video diffusion model (VDM) with trainable parameters $\theta$. Its architecture (Figure 2) builds upon the video transformer diffusion model proposed in Fashion-VDM (Karras et al., 2024). However, it does not include any person image representation and the driving poses are encoded in both 2D and 3D. Our VDM additionally implements two identical sets of temporal blocks to separately handle video motion and 3D spin motion. We describe these adaptations in further detail below. Additional architecture and implementation details are provided in the appendix.

**Garment and Pose Conditioning:** Given a noisy video $z_t$ at diffusion timestep $t$, a UNet encoder $\mathcal{E}_z$ encodes $z_t$ into features $f_z$. Similarly, separate UNet encoders encode the input garment $I_g$ and

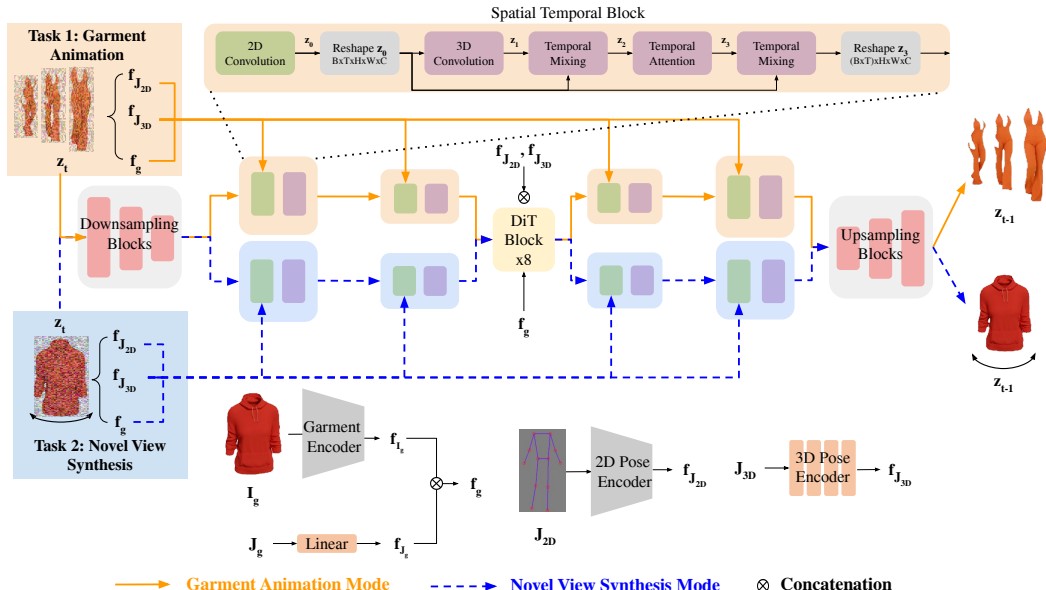

Figure 2: **Architecture.** Our VDM generates video frames conditioned on a garment image $I_g$ and driving pose sequence $(J_{2D}, J_{3D})$. By training separate sets of temporal blocks for each training task (orange for animation and blue for NVS), our VDM can effectively generate either dynamic or spin motion. Refer to Section 4.1 for details.

driving 2D poses $J_{2D}$ into $f_{I_g} = \mathcal{E}_{I_g}(I_g), f_{J_{2D}} = \mathcal{E}_{J_{2D}}(J_{2D})$. Meanwhile, the garment pose $J_g$ and 3D driving pose sequence $J_{3D}$ are encoded into $f_{J_g} = \mathcal{E}_{J_g}(J_g), f_{J_{3D}} = \mathcal{E}_{J_{3D}}(J_{3D})$ by a single linear and 4 dense layers, respectively. Garment and pose features are subsequently concatenated: $f_g = f_{I_g} \oplus f_{J_g}$ and $f_j = f_{J_{2D}} \oplus f_{J_{3D}}$. Mathematically, at timestep $t$, the VDM performs one denoising step to recover noise $\hat{\epsilon}_t$.

$$\hat{\epsilon}_t = \text{VDM}_\theta(z_t, t, f_g, f_j) \tag{2}$$

Inside the VDM, these conditional features are processed in the VDM via transformer blocks (DiT (Peebles & Xie, 2022)). Pose features $f_j$ are spatially-aligned with $f_z$, so they are concatenated channel-wise before self-attention. Meanwhile, the non-spatially aligned garment features $f_g$ and noisy video features $f_z$ are cross-attended. In this manner, the garment features are implicitly warped to their target locations according to the driving poses (Zhu et al., 2023).

**Disjoint Temporal Blocks:** To effectively disentangle dynamic and static spin motions, we implement our VDM with two identical, disjoint sets of temporal blocks. Each set consists of 3D-convolution, temporal attention, and temporal mixing blocks (Blattmann et al., 2023). One set is trained exclusively on batches of *real*-world images and videos $\mathcal{D}_{\text{real}}$, while the other is trained only on batches of static *spin* renderings of 3D garment assets $\mathcal{D}_{\text{spin}}$. This architecture allows the VDM to better synthesize either dynamic or static spin motion, depending on the which set of temporal blocks are activated.

## 4.2 IMPLICIT TRAINING WITH REAL AND SYNTHETIC DATA

We introduce a novel implicit training paradigm to learn photorealistic garment NVS without real-world paired data. Implicit training leverages two or more related tasks for jointly training a model to perform a desired task. In contrast to *joint* training (Ho et al., 2022), where ground-truth task-specific data is supplemented with similar, non-task-specific data, an *implicit* training strategy leverages solely non-task-specific data to learn the target task.

In this case, the desired task is 3D-consistent NVS from real-world garment images, and the training tasks are (1) garment animation using real image and video data $\mathcal{D}_{\text{real}}$ and (2) NVS using synthetic 3D data $\mathcal{D}_{\text{spin}}$. Garment animation with real data trains the model to handle the desired *input style* – diverse, real-world garments, even under challenging conditions, like occlusions and wrinkling.

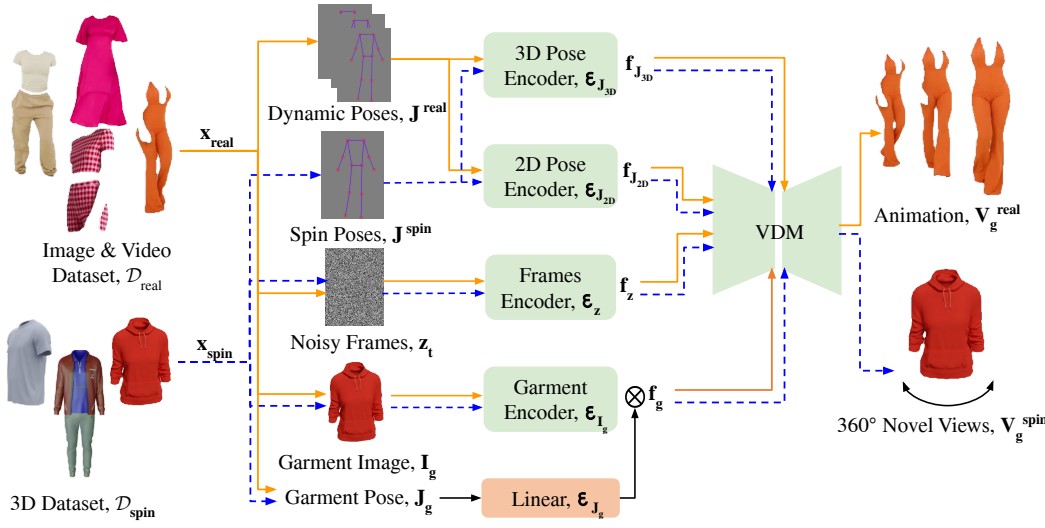

**Figure 3: Implicit Training Paradigm** The VDM is trained to generate either a garment animation given dynamic driving poses (solid orange path) or 360° novel views given static spin driving poses (blue dotted path). By training on both real 2D data and synthetic 3D data, the VDM implicitly learns canonical 360° NVS from real-world garment images, without paired real-world 3D data.

NVS with synthetic 3D data trains the model to generate the desired *output style* – unoccluded, static 360° views (spin videos) of garments. As shown in Figure 3, we train our VDM by alternating batches $x_*$ from both datasets:

$$x_{\text{real}} = (V_g^{\text{real}}, I_g^{\text{real}}, J_g, J^{\text{real}}) \sim \mathcal{D}_{\text{real}} \tag{3}$$

$$x_{\text{spin}} = (V_g^{\text{spin}}, I_g^{\text{spin}}, J_g, J^{\text{spin}}) \sim \mathcal{D}_{\text{spin}} \tag{4}$$

During training, the VDM trains different temporal parameters for handling dynamic motion and static spin motion. For dynamic batches $x_{\text{real}}$, the VDM operates with $\theta_{\text{real}}$ and for spin batches $x_{\text{spin}}$, the VDM operates with $\theta_{\text{spin}}$ (Section 4.1). In this way, disjoint sets of temporal blocks are separately optimized for the different motion styles. After encoding the conditional inputs,

$$\hat{\epsilon}_t = \begin{cases} \text{VDM}_{\theta_{\text{real}}}(z_t, t, f_g^{\text{real}}, f_j^{\text{real}}) & x_{\text{real}} \sim \mathcal{D}_{\text{real}} \\ \text{VDM}_{\theta_{\text{spin}}}(z_t, t, f_g^{\text{spin}}, f_j^{\text{spin}}) & x_{\text{spin}} \sim \mathcal{D}_{\text{spin}} \end{cases} \tag{5}$$

Recall from Section 4.1, $f_g^{\text{real}} = \mathcal{E}_g(I_g^{\text{real}})$ and $f_g^{\text{spin}} = \mathcal{E}_g(I_g^{\text{spin}})$. Let $F_G$ be the garment encoder's ($\mathcal{E}_g$) embedding space for all garments $g \sim G$. Then,

$$f_g^{\text{real}}, f_g^{\text{spin}} \sim F_G \tag{6}$$

Thus, real and synthetic garment embeddings share an embedding space $F_G$, which is optimized for both sets of model parameters ($\theta_{\text{real}}$ or $\theta_{\text{spin}}$). This implies that both input garment styles (real and synthetic) are compatible with both output motion styles (dynamic or static spin). Critically, this property enables HoloGarment to mix and match input garment styles with output motion styles.

## 4.3 REAL-WORLD GARMENT IMAGE(S)-TO-360° GARMENT

Our implicit training approach enables us to train a robust garment embedding space $F_G$ on diverse, large-scale garment video data that is also compatible with the NVS task. As a result, we can accomplish the desired implicit task of *real* image-to-360° NVS. Given a real garment image $I_g^{\text{real}}$ (task 1) and static spin pose sequence in a canonical A-pose $J^{\text{spin}}$ (task 2), HoloGarment generates static 360° novel views of the input garment. During inference, the VDM operates with parameters $\theta_{\text{spin}}$ and denoises noisy frames $z_t$ via iterative noise prediction:

$$\hat{\epsilon}_t = \text{VDM}_{\theta_{\text{spin}}}(z_t, t, f_g^{\text{real}}, f_j^{\text{spin}}) \tag{7}$$

---

**Algorithm 1:** Garment Atlas Finetuning on a Dynamic Garment Video

---

**Input:** $V_g^{\text{real}}$: Input dynamic garment video
**Output:** $f_{\text{atlas}}$: The finetuned garment embedding.
**Initialize:**
  • Freeze all parameters $\theta$
  • $f_{\text{atlas}} \leftarrow$ random embedding of shape $f_g$
**for** $i = 1$ *to* $M$ **do**
  Sample frames $v_g \sim V_g^{\text{real}}$
  Sample timestep $t \sim \text{Uniform}(1, T)$
  Sample noise $\epsilon_t \sim \text{Gaussian}(0, I)$
  Compute poses $(J_{2D}^{\text{real}}, J_{3D}^{\text{real}})$ from $v_g$
  Compute pose embeddings $f_j^{\text{real}}$
  $z_t \leftarrow \text{AddNoise}(v_g, \epsilon_t, t)$                     ▷ Get noisy frames
  $\hat{\epsilon}_t \leftarrow \text{VDM}_{\theta_{\text{real}}}(z_t, t, f_{\text{atlas}}, f_j^{\text{real}})$      ▷ Predict noise using VDM
  $\mathcal{L} = \|\hat{\epsilon}_t - \epsilon_t\|_2^2$               ▷ Compute MSE loss
  $f_{\text{atlas}} \leftarrow \text{Update}(f_{\text{atlas}}, \mathcal{L})$            ▷ Update garment atlas
**end**

---

Since $f_g^{\text{real}} \sim F_G$ (Eq. 6) is compatible with both $\theta_{\text{real}}$ and $\theta_{\text{spin}}$, $\text{VDM}_{\theta_{\text{spin}}}$ generates consistent novel views of the real-world input garment style of task 1 in the output motion style of task 2: static, a-posed, and without occlusions, deformations, or wrinkling.

## 4.4 VIDEO-TO-360° GARMENT VIA ATLAS FINETUNING

As a result of our implicit training approach (4.2), the VDM's garment embedding space $F_G$ is shared between the 2D video animation and 3D NVS tasks. This enables another capability: finetuning a single garment embedding $f_g \sim F_G$ on a dynamic video (task 1) to run NVS (task 2). We call this finetuned garment-specific embedding the garment "atlas", $f_{\text{atlas}}$, which can be leveraged for both tasks (Eq. 6). Note that this is in contrast with earlier subject finetuning methods (Ruiz et al., 2023), which finetune a model on the same task (text-to-image) as during inference (text-to-image).

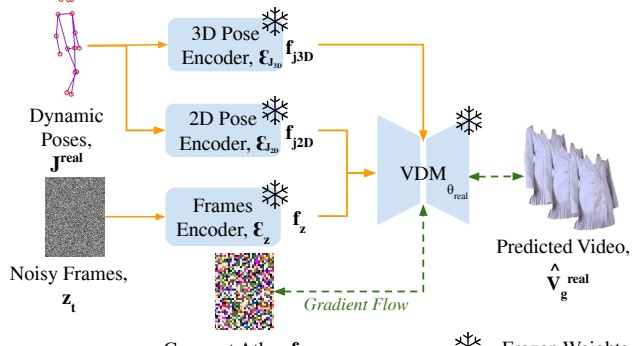

Figure 4: **Garment Atlas Finetuning** HoloGarment enables video-to-NVS by finetuning a garment-specific embedding, or "atlas", on a real-world video. By utilizing this atlas during inference, HoloGarment generates photorealistic 360° novel views of the garment.

The garment atlas finetuning strategy is shown in Figure 4. Initially, $f_{\text{atlas}}$ is randomly initialized with same the shape as $f_g$. Then, freezing all other model parameters $\theta_{\text{real}}$, $f_{\text{atlas}}$ is finetuned on a specific garment video $V_g$ for $M$ iterations. Refer to Algorithm 1 for details. At inference, $f_{\text{atlas}}$ replaces the original garment embeddings $f_g$,

$$\hat{\epsilon}_t = \text{VDM}_{\theta_{\text{spin}}}(z_t, t, f_{\text{atlas}}, f_j^{\text{spin}}) \tag{8}$$

## 5 EXPERIMENTS

### 5.1 DATASETS

We train our model on a combination of real-world and synthetic data. Our real-world dataset includes 17M crawled garment images and 52K garment videos, along with the UBC Fashion Video dataset (Zablotskaia et al., 2019), containing 500 train and 100 test videos. For synthetic data, we use 8,473 unique garment assets, from turboquid (tur), objaverse (Deitke et al., 2022), and other

| Method | Custom Dataset | | | | UBC Dataset | | | |
|---|---|---|---|---|---|---|---|---|
| | FID ↓ | CLIP ↑ | FVD ↓ | SSIM ↑ | FID ↓ | CLIP ↑ | FVD ↓ | SSIM ↑ |
| Garment3DGen | 320 | 0.631 | 2477 | 0.002 | 310 | 0.638 | 2534 | 1.72e-5 |
| Gemini 2.5 Flash Image[1] | 156 | 0.836 | – | 0.700 | 137 | 0.880 | – | 0.742 |
| SVC | 130 | 0.855 | 1109 | 0.767 | 200 | 0.835 | 1144 | 0.745 |
| CAT3D | 152 | 0.861 | 1073 | 0.578 | 186 | 0.798 | 1137 | 0.625 |
| ours$_{video}$ | 131 | **0.890** | 1088 | 0.730 | 128 | 0.872 | 1053 | 0.743 |
| ours$_{3D}$ | 144 | 0.871 | 968 | 0.739 | 147 | 0.847 | **880** | 0.754 |
| **ours** | **128** | 0.872 | **875** | **0.729** | **127** | **0.881** | **880** | **0.771** |

Table 1: **Quantitative Comparisons.** HoloGarment outperforms related and ablated methods on all metrics. [1]Given that the Gemini 2.5 Flash Image model isn't designed for temporal consistency, we chose to omit the FVD metric from our evaluation.

online sources. For each 3D garment, we render 32 views covering one full 360° orbit around the object center. Each image and video frame $I$ is preprocessed using an in-house equivalent of Graphonomy (Gong et al., 2019) to compute the corresponding 2D person keypoints $J_{2D}$, garment segmentation $I_g$, and 2D pose of the garment image $J_g$. Each 2D pose $J_{2D}$ is further preprocessed as a heat-map representation that is spatially-aligned with $I$. Mediapipe (Lugaresi et al., 2019) is used to compute 3D person keypoints $J_{3D}$. During evaluation, each input pair consists of 1 or 3 real-world images or frames form a held-out dataset similar to (1) and a randomly selected pose sequence from (2), covering one full 360° spin.

## 5.2 EVALUATION METRICS

We evaluate our method based on garment fidelity, multi-view consistency, and 3D realism. Garment fidelity is measured using the Fréchet Inception Distance FID (Heusel et al., 2017) and CLIP (Radford et al., 2021) scores between the input garment images and the predicted garment images. For multi-view consistency and 3D realism, we compute the Fréchet Video Distance (FVD) (Unterthiner et al., 2018) and structural similarity (SSIM) between our generated frames and ground-truth rendered 360° spin videos.

## 5.3 360° NOVEL VIEW SYNTHESIS OF IN-THE-WILD GARMENTS

**Image-to-360°:** Figure 6 showcases qualitative results from our image-to-360° NVS method. HoloGarment synthesizes consistent and realistic novel views in a canonical target pose from single-view (top row) or multi-view inputs (middle row), even in the presence of occlusions, pose changes, and wrinkling in the input images. Moreover, HoloGarment handles a variety of garments, including tops, dresses, jackets, rompers, and pants.

**Video-to-360°:** To synthesize 360° novel views of a garment in a dynamic video, we finetune a garment embedding, which we call a garment "atlas" $f_{atlas}$, on a 128-frame real-world video for 500 iterations with batch size 32 and constant learning rate of $1e-3$. By only optimizing for $f_{atlas}$, we prevent undesired overfitting to the input video motion. We showcase qualitative examples of our video-to-360° NVS method in the bottom row of Figure 6 and Figure 9 of the appendix. Our atlas finetuning strategy enables HoloGarment to consolidate an arbitrary number garment views, poses, and deformations into a unified 360° garment representation.

## 5.4 COMPARISONS TO STATE-OF-THE-ART

We compare our method to image-to-3D (Garment3DGen (Sarafianos et al., 2024) and CAT3D (Gao et al., 2024)), image editing (Gemini 2.5 Flash Image (Google, 2025)), and camera-controlled video generation (Stable Virtual Camera (Zhou et al., 2025), Veo3 (Google DeepMind, 2024)) methods. Our results are presented in Figure 5 and Table 1. Qualitatively, HoloGarment produces superior results compared to Garment3DGen, CAT3D, and Stable Virtual Camera, and achieves visual quality on par with large, publicly available models like Gemini 2.5 Flash Image and Veo3. This is notable given that our model was trained on a significantly smaller dataset. Quantitatively, our approach consistently outperforms all compared methods across all metrics on both evaluation datasets (Table 1). Due to the high computational cost of large-scale evaluation, a full quantitative comparison with

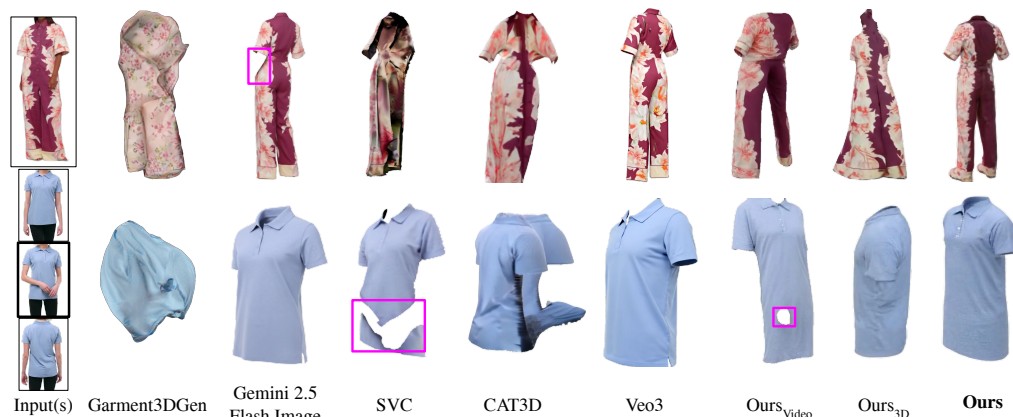

Figure 5: **Qualitative Comparisons.** HoloGarment demonstrates superior preservation of garment appearance, realism, canonical pose, and multiview consistency, as well as robustness to occlusions, compared to related works and the ablated versions.

Veo3 is reserved for future work. Further discussion and implementation details for each method are provided in the appendix.

## 5.5 ABLATIONS

**Implicit Training Paradigm:** As shown in Figure 5 and Table 1, our implicit video-and-3D training approach (ours) outperforms video-only training (ours$_{video}$) and synthetic 3D-only training (ours$_{3D}$) methods. Video-only training leads to the great photorealism and garment fidelity, but it fails to enable 3D-consistent motion generation or realistic novel views. Plus, the synthesized views contain holes where limbs or hair overlap with the garment in the input views. 3D-only training generates highly consistent and realistic garment spins, but exhibits poor garment fidelity and over-smoothed textures. Our jointly-trained model balances the benefits of video-only and 3D-only training, maintaining garment fidelity, photorealism, and multi-view consistency without holes.

**Atlas Finetuning:** We evaluate our atlas finetuning strategy quantitatively in Table 2 and qualitatively in the appendix. Atlas finetuning enables our model to consolidate information from an arbitrary number of images, improving fidelity and realism. Compared to single-image conditioning, atlas finetuning improves performance across all metrics.

| Method | FID ↓ | CLIP ↑ | FVD ↓ | SSIM ↑ |
|---|---|---|---|---|
| w/o Atlas | 134 | 0.852 | 538 | 0.689 |
| w/ Atlas | **103** | **0.900** | **474** | **0.728** |

Table 2: **Atlas Finetuning Ablation.** We evaluate 40 results with and without atlas finetuning.

## 6 DISCUSSION

In this paper, we present HoloGarment, a method for synthesizing state-of-the-art novel views of garments in real-world images and videos. We introduce an implicit training scheme to optimize a video diffusion model for real-world garment image-to-360° novel-view synthesis (NVS) using a combination of large-scale 2D garment data and limited synthetic 3D garment assets. We further propose atlas finetuning, a strategy where a garment embedding, or "atlas", is finetuned on a dynamic garment video to enable video-to-NVS capabilities.

**Limitations:** While our method improves over existing methods, it faces several limitations. Due to the limited diversity of the synthetic 3D garment dataset, HoloGarment struggles with unusual garment shapes (e.g. assymetry or cut-outs).See the supplementary for qualitative examples. A larger synthetic garment dataset may remedy such issues. Other future work includes speeding up atlas finetuning (currently ∼30 minutes on a single TPU) and increasing resolution via super-resolution network.

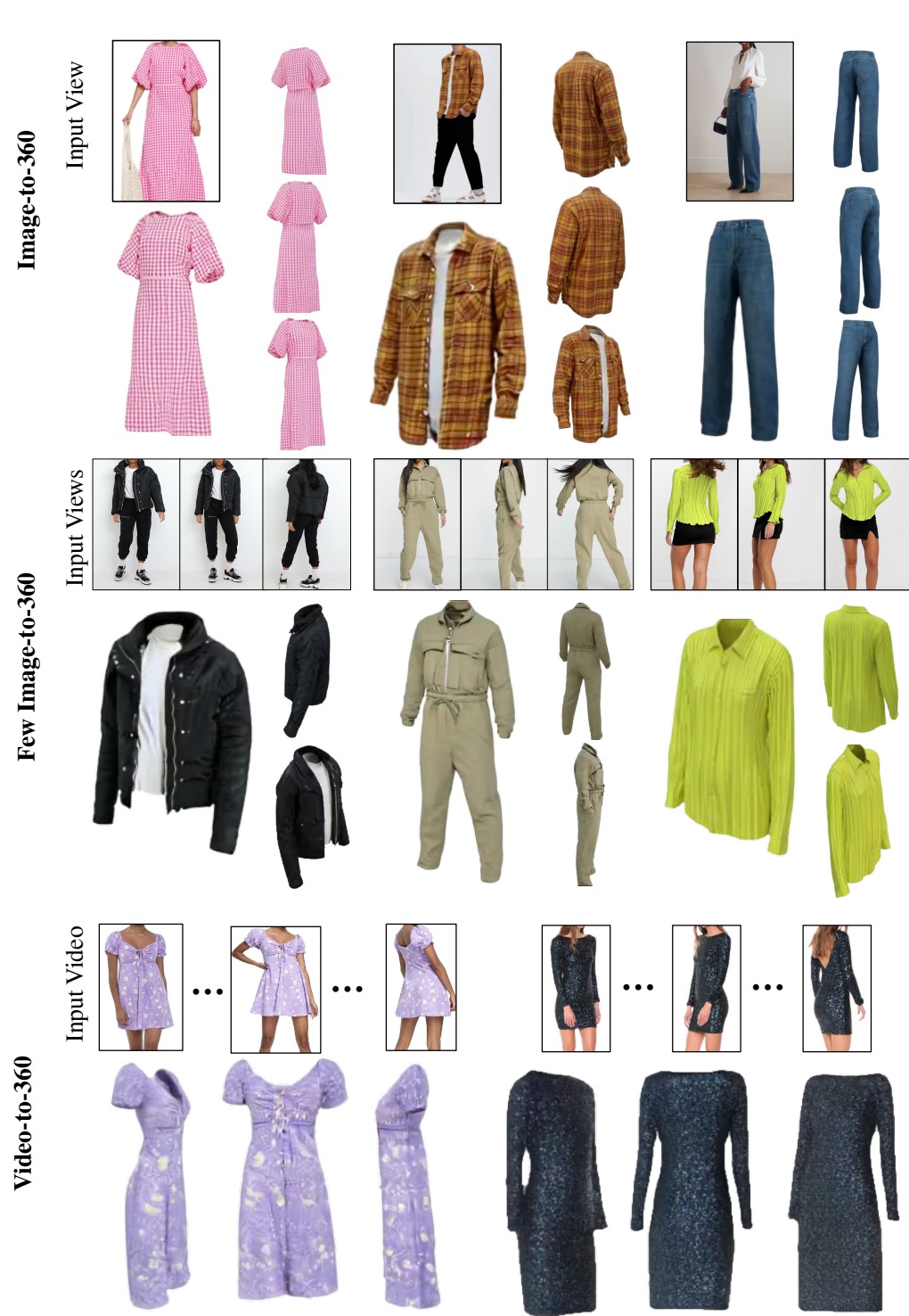

Figure 6: **Qualitative Results.** Our method generates 360° novel views of garments from single images, multiple images, or videos. Additional qualitative results are shown in the supplementary material.

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
