# OpenReview forum: "HoloGarment: 360$\degree$ Novel View Synthesis of In-the-Wild Garments"
_ICLR.cc/2026/Conference — Submitted to ICLR 2026_

### Official Review · Reviewer_71FH · 2025-10-25

**Soundness:** 3
**Presentation:** 3
**Contribution:** 3
**Rating:** 6
**Confidence:** 4

**Summary:**

This paper presents a method to generate novel view image of a given garment.
The novelty lies in the use of a video diffusion model and real world video data to learn a shared garment embedding space.
The quality of the results demonstrated in the paper are better compared to previous works.

**Strengths:**

- The method cleverly leverage real-world garment videos and video generation task to enable multi-view garment generation without paired data training.
- The quality of the generated results are better than previous methods.
- The exposition of the paper is clear and the idea is not difficult to follow.

**Weaknesses:**

- While the method works nicely, the additional training effort might be large. And the computational requirements and performances are not explicitly discussed in the main paper.
- Regarding evaluation, I think it is better to have a small testing set with ground truth for easier comparison. For example, one can record a video of a person wearing a given garment and perform some actions. In this case, we can compare the reconstructed garment more explicitly.
- The comparison with methods that have explicit geometry, e.g., Garment3DGen, is a bit weird since the proposed method does not have a mesh geometry. That means, it is difficult to edit the result garments unlike those method with explicit geometry. I would suggest at least to mention the advantage of those methods.

**Questions:**

- The internal color and structure seems not inferred correctly, what might be the main reason for this? (see results in Fig. 6)
- What are the possibilities of enhancing symmetry, e.g., for results in Fig. 6?
- While the goal of the generated garment seems to the garment without wrinkle, but given some human body poses, the generated garment without wrinkles seem unnatural. I am curious about whether the goal is reasonable?

---

> ### Author Response · Authors · 2025-11-19
> **Authors' Rebuttal**
>
> To reviewer 71FH,
>
> Thank you for sharing your feedback and concerns with us. We appreciate your input and would like to address your points with the following clarifications.
>
> (1) *“While the method works nicely, the additional training effort might be large. And the computational requirements and performances are not explicitly discussed in the main paper.”*
>
> We provide training and inference time and compute requirements in Supplementary Section B.
>
> (2) *“Regarding evaluation, I think it is better to have a small testing set with ground truth for easier comparison. For example, one can record a video of a person wearing a given garment and perform some actions. In this case, we can compare the reconstructed garment more explicitly.”*
>
> Our target output is a 360-degree spin video of a static, a-posed, standalone garment without occlusions and without a person wearing the garment. Collecting such real-world videos would be extremely challenging. However, we do provide a comparison of our method with ground-truth rendered views of 3d garment assets in Supplementary Figure 7.
>
> (3) *“The comparison with methods that have explicit geometry, e.g., Garment3DGen, is a bit weird since the proposed method does not have a mesh geometry. That means, it is difficult to edit the result garments unlike those method with explicit geometry. I would suggest at least to mention the advantage of those methods.”*
>
> Following recent trends of 3d asset generation (e.g. DreamFusion, MVDream, Cat3D), reconstruction of meshes from synthesized multi-view frames is a well-adapted paradigm, which works quite well combined with our method in practice. As such, novel view synthesis and 3D reconstruction are closely related tasks. However, we will mention the benefits of methods that construct explicit geometry for downstream tasks, such as editing.
>
> (4) *"The internal color and structure seems not inferred correctly, what might be the main reason for this? (see results in Fig. 6)"*
>
> The garment’s geometry is learned from synthetic 3D garments, which frequently do not contain internal textures, leading to gray internal appearance during inference. We will leave this to future work.
>
> (5) *What are the possibilities of enhancing symmetry, e.g., for results in Fig. 6?*
>
> In this work, we focus on reconstructing the garment in a canonical pose without occlusion. If the input image shows the symmetrical structure of the garment, the model would respect that in theory. However, for hallucinated areas under occlusion, increasing the size of the 3d garment dataset would likely also increase symmetry.
>
>
> (6) *"While the goal of the generated garment seems to the garment without wrinkle, but given some human body poses, the generated garment without wrinkles seem unnatural. I am curious about whether the goal is reasonable?"*
>
> Similar to question 5, our goal is to generate a canonical view of the garment in a fixed a-pose – free of wrinkles for better, more-informative garment visualization. At the same time, certain folds should remain in the output because they are part of the garment’s inherent design rather than pose-dependent artifacts. For example, pleated skirts (supplementary video 0:05-0:09) and ruched dresses (supplementary video 0:10-0:16) naturally exhibit persistent structure that is independent of posture; in such cases, preserving those folds is the correct behavior. Our method does so consistently. Furthermore, future work may reconstruct 3D canonical mesh from our novel views using a 3D-reconstruction method, which requires such a canonical pose.

---

> > ### Comment · Reviewer_71FH · 2025-11-20
> >
> > thanks authors for the rebuttal, most of my questions are addressed.

---

### Official Review · Reviewer_JQuA · 2025-10-31

**Soundness:** 2
**Presentation:** 2
**Contribution:** 2
**Rating:** 2
**Confidence:** 4

**Summary:**

Taking 1-3 garment images or a video as input, this work aims to output its 360-degree novel view synthesis. To train such a model usually requires a large amount of data. However, 360-degree garment images are very hard to collect. And, to use synthetic data will also cause significant domain gap. This work considers to use large-scale real video data (with no 360-degree but owns diverse deformation and short-period view changing) and small-scale synthetic 3D data. Based on this, an implicit training mechanism is proposed. The key idea is to learn a shared garment embedding space between both domains.

**Strengths:**

- the motivation is clear and the target problem of briging real and synthetic gap is significant.

**Weaknesses:**

- In all, I think the experiments and analysis are not sufficient. The key contribution is the training paradigm, but it lacks in-depth analysis about why it works. There are two tasks, one is novel pose synthesis for real data and another one is novel view synthesis for 3D data, how do these two tasks benefit each other?

- In Fig 5, it seems Ours_3d produces very bad quality. Is it because the limited data scale (only ~8000)? or the real-sim domain gap? It is unclear. I think it is not hard to have more synthetic 3D garment assets, for example, we can use image-to-3D models to do generation from images.

- The visual comparisons are limited, only two examples are shown in Fig 5 and the result of the proposed method for the first example is not aligned with the input. This makes me doubt on the generalization ability of the proposed approach.

- Another primary concern is from the limited contribution. It seems the key contribution is just a training strategy. At first, I really do not know why it is called "implicit". And, it is also hard to see if the proposed training strategy can be used for other tasks.

**Questions:**

No.

---

> ### Author Response · Authors · 2025-11-19
> **Authors' Rebuttal (Part 1)**
>
> To reviewer JQuA,
>
> Thank you for sharing your feedback and concerns with us. We appreciate your input and would like to address your points with the following clarifications.
>
> (1) *"Another primary concern is from the limited contribution… And, it is also hard to see if the proposed training strategy can be used for other tasks.”*
>
> We would like to clarify that **HoloGarment is the first method to address the significantly harder task of real-world garment 360° NVS**, where inputs exhibit occlusion, pose variation, nonrigid deformation, and real-world lighting—conditions under which existing object-centric NVS methods are not designed to operate. **Our primary contribution is not a marginal improvement in quality, but enabling a previously infeasible task**—real-world garment NVS from casually captured images and video.
>
> As shown in Table 1, Figure 5, and the supplementary video (1:16–1:56), HoloGarment consistently outperforms CAT3D and Garment3DGen in both fidelity, photorealism, and view consistency. These methods struggle substantially when faced with occlusions and pose-induced deformation, whereas our implicit training scheme remains robust.
>
> HoloGarment’s contribution opens a new research direction for NVS under real-world constraints, with **applications far beyond garments**—e.g., reconstructing objects partially hidden in vegetation or clutter, or recovering occluded objects in aerial or satellite imagery. The technical insights (implicit 2D–3D training, split temporal blocks, atlas finetuning) are broadly applicable to any scenario where full 3D capture is impractical.
>
>
> (2) *“It seems the key contribution is the training strategy. At first, I really do not know why it is called ‘implicit’."*
>
> The primary motivation for our implicit training paradigm is the lack of paired real-world data (real image/video ←→ 3d garment asset) (lines 208–212). Real-world videos contain occlusions, wrinkles, and body-induced deformations that do not exist in synthetic 3D renders, so naïvely generating synthetic “paired” data is not a valid substitute. Implicit training is designed specifically to overcome this fundamental mismatch.
>
> The training is called “*implicit*” because the model learns the target task indirectly through related tasks—namely, real-world video animation and synthetic 3D novel-view synthesis—rather than being provided explicit ground-truth supervision for the target real-world 360° NVS task. This is in contrast to *explicit training,* where paired input-output supervision is available for the target task.
>
>
> (3) *"The key contribution is the training paradigm, but it lacks in-depth analysis about why it works. There are two tasks, one is novel pose synthesis for real data and another one is novel view synthesis for 3D data, how do these two tasks benefit each other?"*
>
> In Section 4.2-4.4, we provide intuition and analysis of how implicit training works, but we will add a conceptual explanation and empirical justification to Section 5.5 to make the intuition behind the training paradigm clearer.
>
> The two tasks—(1) pose-guided image/video generation from real videos and (2) novel-view synthesis from 3D assets—provide **complementary supervisory signals** to jointly train our model for real-world 360° garment NVS.
>
> These two tasks address opposite weaknesses: (1) Real videos teach the **input appearance space**, but lack complete 360° supervision. (2) Synthetic 3D data teach the **output viewpoint space**, but lack real-world artifacts.
>
> **Real-video supervision (Task 1)** exposes the model to a real distribution of images containing occlusions, wrinkles, lighting, pose changes, and nonrigid deformations, factors that never appear in 3D renderings. This branch teaches the *shared garment encoder* how to handle real garments with challenging imaging artifacts and provides robustness to diverse, in-the-wild garment types. Plus, it trains the decoder to  generate photorealistic output images.
>
> **3D NVS supervision (Task 2)** provides the model with geometrically consistent, unoccluded 360° spins that are rare or impossible to observe in real footage (e.g., canonical, A-posed rotations without occlusions). This branch teaches the *NVS-specific temporal blocks* about global structure, cross-view consistency, and the correct target output distribution for full 360° NVS.
>
> This analysis is quantitatively supported in Table 1, and qualitatively in Figure 5 and the supplementary video (1:57–2:09), where single-task training collapses in either realism (3D-only) or structural consistency (video-only).

---

> ### Author Response · Authors · 2025-11-19
> **Authors' Rebuttal (part 2)**
>
> (4) *“In Fig 5, it seems Ours_3d produces very bad quality. Is it because the limited data scale (only ~8000)? or the real-sim domain gap? It is unclear.”*
>
> Thank you for the question. **Ours_3d** corresponds to our ablation trained only on the limited 3D dataset (8,473 assets); the **full model results are shown in the last column** of Figure 5.
>
> The lower quality of this ablation arises from two factors:
> - **Domain gap**: The 3D assets do not contain real-world artifacts—occlusions, wrinkles, deformations, or diverse poses. Consequently, a model trained solely on 3D data struggles to generalize to real images, producing outputs that are noticeably less photorealistic than the full model.
> -**Limited 3D dataset scale**: With only ~8k assets, the model is insufficiently exposed to diverse garment types. For example, simple, over-represented categories like t-shirts are reconstructed relatively well, but complex garments like jumpsuits are not robustly handled.
>
> (5) *“I think it is not hard to have more synthetic 3D garment assets, for example, we can use image-to-3D models to do generation from images.”*
>
> Similar to (1), using existing image-to-3D methods to generate 3D garment meshes from real-world images–where inputs exhibit occlusion, pose variation, nonrigid deformation, and real-world lighting–is *infeasible*. The motivation of our work is to enable this task.
>
> As shown in Table 1, Figure 5, and the supplementary video (1:16–1:56), HoloGarment consistently outperforms prior image-to-3D methods, which struggle substantially when faced with occlusions and pose-induced deformation.
>
> (6) *“The visual comparisons are limited, only two examples are shown in Fig 5 and the result of the proposed method for the first example is not aligned with the input. This makes me doubt on the generalization ability of the proposed approach.”*
>
> The limited visual comparisons in the main paper are due to space constraints; **we provide additional examples in the supplementary video (1:15–1:56)**, and we plan to include even more comparisons in the final version.
>
> Regarding the first example, while the top input image may be difficult to interpret without zooming in, it depicts a short-sleeved jumpsuit, and **HoloGarment correctly reconstructs the back view of this garment**, demonstrating that the model generalizes to diverse garment types even under challenging poses and occlusions.

---

> > ### Comment · Reviewer_JQuA · 2025-11-28
> > **Thanks for responses.**
> >
> > Many thanks to the authors to provide detailed explanations.
> >
> > First of all, I agree that full-view synthesis without sufficient data is a challenging task. And, I really appreciate your efforts to make the first attempt. However, I have to be honest that your rebuttal still do not convince me for the following reasons.
> > 1) for Q5, image-to-3D I mentioned actually stands for Trellis, Hunyuan models. Yes, it cannot generate clean garment meshes. However, using them to generate 3D full models, segment the garment out and render multi-view images is very easy to do based on my experiences. From this aspect, it is not hard to have a large amount of synthetic data. With sufficient synthetic data, I am very curious about what will happen.
> > 2) I also agree with another reviewers that the results are full of artifacts, blurry appearances from the video. This makes me hard to give a positive evalutaion.

---

### Official Review · Reviewer_HVSd · 2025-10-31

**Soundness:** 3
**Presentation:** 3
**Contribution:** 3
**Rating:** 4
**Confidence:** 4

**Summary:**

The paper proposes a method to create a 360 degree novel view of garment in a canonical pose from few images or videos. Existing method fails in the occlusions or produce noisy rendering while the proposed method is robust. The method uses combination of real 2D data and synthetic 3D data to learn shared garment embedding for novel view synthesis. The results are quite good and the video is satisfactory.

**Strengths:**

The paper uses the Fashion VDM model well; the shared representation os garment seems a good idea.
The implicit training paradigm seems working for this problem quite well and the authors design the architecture elegantly.
The quantitative results are encouraging and the qualitative results are good.

**Weaknesses:**

It is not clear how the occlusion is solved? Which part of the network is responsible to solve this problem? Or is it by data?
Why Spinning is handled differently? Isnt spin a form of dynamic poses?
What is garment pose and how is it estimated?
If the garment has different texture (not symmetry) and they are visible in 1-3 views then how the method perform? Need a visual results for this.

**Questions:**

Please refer weaknesses.

---

> ### Author Response · Authors · 2025-11-19
> **Authors' Rebuttal**
>
> To reviewer HVSd,
>
> Thank you for sharing your feedback and concerns with us. We appreciate your input and would like to address your points with the following clarifications.
>
> (1) *"It is unclear how the occlusion is solved? Which part of the network is responsible to solve this problem? Or is it by data?"*
>
> The key idea is that occlusion is not something the model “solves” explicitly through masking or geometry, but rather something it learns to handle through our **implicit training paradigm**.
>
> In implicit training, the two tasks—(1) pose-guided image/video generation from real videos and (2) novel-view synthesis from 3D assets—provide complementary supervisory signals to jointly train our model for real-world 360° garment NVS. Tasks 1 and 2 address opposite weaknesses: (1) Real videos teach the **input appearance space**, but lack complete 360° supervision. (2) Synthetic 3D data teach the **output viewpoint space**, but lack real-world artifacts.
>
> **Real-video supervision (Task 1)** exposes the model to a real distribution of images containing occlusions, wrinkles, lighting, pose changes, and nonrigid deformations, factors that never appear in 3D renderings. This branch teaches the shared garment encoder how to handle real garments with challenging imaging artifacts and provides robustness to diverse, in-the-wild garment types. Plus, it trains the decoder to  generate photorealistic output images.
>
> **3D NVS supervision (Task 2)** provides the model with *geometrically consistent, unoccluded 360° spins* that are rare or impossible to observe in real footage (e.g., canonical, A-posed rotations without occlusions). This branch teaches the NVS-specific temporal blocks about global structure, cross-view consistency, and the correct target output distribution for full 360° NVS.
>
> Through shared parameters, **the model learns to map occluded, real inputs (Task 1) to complete 360° unoccluded outputs (Task 2)**. In other words, occlusion handling emerges from the complementary supervision: real data teaches how to interpret occluded inputs, while 3D data teaches what the unoccluded target should look like.
>
> We will clarify this in Section 4.2 of the paper.
>
> (2) *"Why spinning is handled differently? Isn’t spin a form of dynamic poses?"*
>
> While “spin” is a form of dynamic poses, it is a **specific motion style** corresponding to Task 2: a canonical, unoccluded 360° rotation. Real videos rarely contain such motion, and when they do, they are heavily occluded and pose-dependent. Therefore, we use distinct temporal blocks:
>
> - **Spin temporal blocks**: Specialized modules that model the clean, deterministic 360° spin motion learned from 3D assets.
> - **Dynamic-motion blocks**: Modules that handle the complex, occlusion-heavy dynamics of real videos.
>
> Separating these motion styles allows the model to keep **real-world robustness** while also producing **geometrically consistent, fully unoccluded 360° spins**, which would not be possible from real-video supervision alone. We empirically find that separating these temporal blocks improves the distinction between the output styles.
>
> We will clarify this in Section 4.1 of the paper.
>
> (3) *"What is garment pose and how is it estimated?"*
>
> Garment pose is computed as the body pose of the human wearing the garment, before the human is segmented out of the garment image.
>
> (4) *“If the garment has different texture (not symmetry) and they are visible in 1-3 views then how the method perform? Need a visual results for this.”*
>
> We show non-symmetric examples  (where the front and back are not identical) in Figure 1, 5, 6, 7, 9, and 11, as well as the supplementary video. If multiple views are provided, then the spin accurately generates the front and back views (see Figure 5 bottom). If only a single view is provided, the model hallucinates the unseen views based on the distribution of training data.

---

> > ### Comment · Reviewer_HVSd · 2025-11-27
> > **Post rebuttal**
> >
> > Based on the answers on occlusion problem and handling on spin motion, I am keeping my original score.

---

### Official Review · Reviewer_xh6o · 2025-11-01

**Soundness:** 2
**Presentation:** 2
**Contribution:** 2
**Rating:** 2
**Confidence:** 5

**Summary:**

The authors in this paper presented a method for synthesizing state-of-the-art novel views of garments in real-world images and videos. An implicit training scheme is introduced to optimize a video diffusion model for real-world garment image-to-360° novel-view synthesis (NVS) using a combination of large-scale 2D garment data and limited synthetic 3D garment assets.

However, the results are not impressive, and missing key experimental setups as discussed below.

**Strengths:**

Key strength of this paper is an implicit training scheme to optimize a video diffusion model for real-world garment image-to-360° novel-view synthesis (NVS) using a combination of large-scale 2D garment data and limited synthetic 3D garment assets.

**Weaknesses:**

- Video results are not impressive, there are many places where texture/pattern continuity across different views is broken. Results on clothes where some textual information is written etc could be a good example to validate this.
- There are many examples where shading effect in the input  images around folds and wrinkles were treated as part of textures and the same is reflected in the generated output which is wrong. Those folds and wrinkles must go away under certain body postures, but it is not happening.
- Results on images where parts of garments are occluded with body parts are limited. Show more video examples with diverse loose garments, not on tight body hugging T-shirts etc.
- Only one failure case is shown, whereas after seeing the supplementary video, I believe there could be more ?
- While on one side author motivated that there method leverages the limited 3D data and that is sufficient. On the contrary in the limitation section authors are saying "larger synthetic garment dataset may remedy such issues", which is contradicting the initial motivation.
- Speed is another concern, taking 30 minutes is a huge. Authors should add a detailed inference and training time analysis /profiling in the paper. Right now this information is missing in the paper.

**Questions:**

please see the weakness section

---

> ### Author Response · Authors · 2025-11-19
> **Authors' Rebuttal (Part 1)**
>
> To reviewer xh6o,
>
> Thank you for sharing your feedback and concerns with us. We appreciate your input and would like to address your points with the following clarifications.
>
>
> (1) *“Video results are not impressive, there are many places where texture/pattern continuity across different views is broken. Results on clothes where some textual information is written etc could be a good example to validate this.”*
>
> We would like to clarify that **HoloGarment is the first method to address the significantly harder task of real-world garment 360° NVS**, where inputs exhibit occlusion, pose variation, nonrigid deformation, and real-world lighting—conditions under which existing object-centric NVS methods are not designed to operate.
>
> **Our primary contribution is not a marginal improvement in quality, but enabling a previously infeasible task**—real-world garment NVS from casually captured images and video.
>
> As shown in Table 1, Figure 5, and the supplementary video (1:16–1:56), HoloGarment consistently outperforms CAT3D and Garment3DGen. These methods struggle substantially when faced with occlusions and pose-induced deformation.
>
> We believe that our work will motivate future research into real-world NVS in various challenging scenarios, beyond garments. For example, NVS of objects in heavily occluded imagery, such as in forested areas or satellite imagery.
>
>
> (2) *"While on one side author motivated that there method leverages the limited 3D data and that is sufficient. On the contrary in the limitation section authors are saying "larger synthetic garment dataset may remedy such issues", which is contradicting the initial motivation."*
>
> This point is important, and we apologize if the motivation was not phrased clearly. There is **no contradiction** between leveraging limited 3D data and noting that a larger synthetic set could further improve performance, because these statements refer to **two different limitations**:
>
> **1. Lack of paired real-world data (real image/video ↔ exact 3D garment).**
>  This is the primary motivation for our implicit training paradigm (lines 208–212). Real-world videos contain occlusions, wrinkles, and body-induced deformations that do not exist in synthetic 3D renders, so naïvely generating synthetic “paired” data is not a valid substitute. Implicit training is designed specifically to overcome this fundamental mismatch.
>
> **2. Limited diversity in available 3D garment assets.**
>  Existing 3D datasets tend to be small and stylized (“avatar-ish”), but they remain valuable for learning geometric priors—for example, how an unoccluded garment should look from canonical views. Because this geometric supervision is complementary to the real-video supervision, more diverse 3D assets would further strengthen this part of the training signal.
>
> In other words, **3D data alone is not sufficient** to learn real-world appearance, but **more 3D data is still beneficial** for improving geometric coverage. Additional 3D assets would enhance performance, but they would not eliminate the need for implicit training, which is required precisely because paired real data does not exist.
>
> We will clarify this distinction in the revision.
>
>
> (3) *"Results on images where parts of garments are occluded with body parts are limited."*
>
> Our evaluation **does include 28 cases where the garment is occluded by arms, hands, and hair**, across both the main paper and supplementary material:
> - **13 in the main paper** (Figures 1, 2, and 6), including cases where arms or hands cover large regions of shirts, dresses, skirts, and jumpsuits.
> - **3 in the appendix** (Figure 11), showing additional severe occlusion scenarios.
> - **15 in the supplementary video**, including sequences where the garment is occluded in every input frame.
>
> Across these examples, HoloGarment reconstructs coherent 360° views despite substantial missing regions in the input, demonstrating that the implicit training paradigm effectively handles occlusions.
>
> (4) *"Show more video examples with diverse loose garments, not on tight body hugging T-shirts etc."*
>
> Our current submission includes **6 cases of not tight, body-hugging garments**:
>
> - 2 in our main paper (Figure 1 and 6), including an oversized shirt and flowy dress
> - 2 in our appendix (Figure 11), including a loose-fitting purple shirt and baggy navy jacket.
> - 2 in our supplementary video; a loose shirt example (0:36-0:40) and loose sweatshirt (1:29-1:32).

---

> ### Author Response · Authors · 2025-11-19
> **Authors' Rebuttal (Part 2)**
>
> (5) *"There are many examples where shading effect in the input images around folds and wrinkles must go away under certain body postures, but it is not happening."*
>
> Thank you for highlighting this issue. While our model does not remove all folds and wrinkles in the generated spin, the primary goal of our method is to correct **large, pose-dependent deformations** that obscure the underlying garment appearance in the input image(s) or video. As shown in our multi-view examples, substantial folds caused by arm raising, torso twisting, or fabric stretching are largely removed in the generated 360° spin, producing a canonicalized, neutral-pose version of the garment.
>
> At the same time, **certain folds should remain in the output** because they are part of the garment’s inherent design rather than pose-dependent artifacts. For example, pleated skirts (supplementary video 0:05-0:09) and ruched dresses (supplementary video 0:10-0:16) naturally exhibit persistent structure that is independent of posture; in such cases, preserving those folds is the correct behavior. Our method does so consistently.
>
> We will revise the paper to explicitly distinguish between pose-induced deformations that should be removed and design-dependent folds that should be preserved, and to better discuss the current limitations of disentangling fine wrinkles from texture.
>
> (6) *"Only one failure case is shown, whereas after seeing the supplementary video, I believe there could be more ?"*
>
> Our submission includes **4 explicit failure-case examples**—two in the supplementary video (2:09–2:14) and two in Appendix Figure 10—along with a detailed discussion in Appendix Section G describing the typical conditions under which failures occur. We will make sure to expand the breadth of failure analysis to also include other failure modes, such as input segmentation failures and light wrinkling appearing in the final spin.
>
> (7) *“Speed is another concern, taking 30 minutes is a huge. Authors should add a detailed inference and training time analysis /profiling in the paper. Right now this information is missing in the paper.”*
>
> Compared to single- or sparse-frame conditioning, finetuning a garment atlas on a specific video improves the quality of video-to-360 NVS, but *is not necessary to achieve good-quality results* (see appendix Figure 9). While 30 minutes is a time-consuming, this is only for the finetuning stage. Inference itself only takes ~3 minutes.
>
> The current submission contains training and inference time details in Appendix Section B, but we will add the inference time in the final version.

---

> > ### Comment · Reviewer_xh6o · 2025-11-28
> > **Post rebuttal**
> >
> > Thanks for the rebuttal, some of the questions are answered but that can push my final rating to max 4.

---

### Meta-Review · Area_Chair_bgRW · 2026-01-01

**Summary:**

Most reviewers recommend rejection, except for Reviewer 71FH.

The AC identifies key concerns: 1) mediocre synthesis quality with blurry and view-inconsistent texture patterns; 2) missing key technical details, ablations, and references, indicating the paper needs further proofreading; 3) a 30-minute inference time is still less competitive than feed-forward methods, lacking superior quality, making the lengthy fine-tuning and optimization process less appealing; and 4) testing examples are mostly solid-colored or feature repeated patterns. More challenging examples, like textual logos or overly loose outfits, could better reveal the true potential of this training paradigm, benefiting from the diverse range of large-scale internet fashion videos.

Therefore, the AC believes this paper falls short of ICLR standards and suggests rejection while urging the authors to address this feedback in future submissions.

**Reviewer Concerns:**

- Reviewer xh6o: Some concerns are addressed, but the rebuttal on "low-quality results" is unconvincing, particularly regarding view-inconsistent and blurry texture patterns, poorer synthesis quality compared to Veo3 (lacking quantitative comparison), and slow inference speed (3 minutes, even tuning-free, is still too slow compared to feed-forward models).

- Reviewer HVSd: The AC shares similar concerns after reviewing the full paper. Key technical details are missing, such as garment pose (2D or 3D? Landmark or SMPL?), and whether the spin pose is deterministic or "randomly selected from D_spin" (L344). What if the spin pose deviates significantly from the body shape of the person in the input images? Necessary analysis should be included. Additionally, the issue of "garments with different textures across views" remains inadequately addressed; while most shown outfits are "non-symmetric," this applies to coarse structure rather than textures. If the front and back views feature different colors and logos, will view consistency be maintained? More visual results should be added to address these questions.

- Reviewer JQuA: The rebuttal does not fully address the concerns. In contrast to "implicit training," it resembles a "semi-supervised" approach that uses strong supervision signals and benefits from a proxy task (novel pose synthesis) with weak labels. 3D paired training data could come from clothed human datasets (CustomHumans, CloSe, 2K2K, THuman2.1, etc.) or be generated via Hunyuan or Trellis and animated using Maximo. JQuA also shares xh6o's concerns about low-quality results and visual artifacts. Regarding novelty, for the NVS/reconstruction task from unconstrained human photos (varied poses, non-rigid deformation, occlusion, etc.), PF-LHM [1], HAVE-FUN [2], PuzzleAvatar [3], and AvatarBooth [4] are relevant works. Although they target full-body clothed human reconstruction, their results could still be compared after proper segmentation and are even more aligned with such an unconstrained setting than Garment3DGen (which uses a single image as input).

- Reviewer 71FH: Concerns have been adequately addressed in the reviewer's reply on November 20.

References:
- [1] Qiu, Lingteng, et al. "PF-LHM: 3D Animatable Avatar Reconstruction from Pose-free Articulated Human Images." arXiv preprint arXiv:2506.13766 (2025).
- [2] Yang, Xihe, et al. "HAVE-FUN: Human avatar reconstruction from few-shot unconstrained images." Proceedings of the IEEE/CVF Conference on Computer Vision and Pattern Recognition. 2024.
- [3] Xiu, Yuliang, et al. "PuzzleAvatar: Assembling 3d avatars from personal albums." ACM Transactions on Graphics (TOG) 43.6 (2024): 1-15.
- [4] Zeng, Yifei, et al. "AvatarBooth: High-quality and customizable 3d human avatar generation." arXiv preprint arXiv:2306.09864 (2023).

**Reviewer Scores:**

- Reviewer xh6o: might raise the score since some concerns are addressed, but still below the acceptance bar (4 $\downarrow$)
- Reviewer HVSd: is very likely to keep the score (4)
- Reviewer JQuA: is very likely to keep the score (2)
- Reviewer 71FH: is very likely to keep the score (6)

---

### Decision · Program_Chairs · 2026-01-26

Reject